# New Inogranic Scintillators' Application in the Electromagnetic Calorimetry in High-Energy Physics

**Dmitry Averyanov** [1,2,*] and **Dmitry Blau** [1,2]

1   National Research Centre "Kurchatov Institute", Moscow 123182, Russia; dmitry.blau@cern.ch
2   Moscow Institute of Physics and Technology, National Research University, Dolgoprudny 141701, Russia
*   Correspondence: dmitry.averyanov@cern.ch

**Abstract:** Scintillation crystals $Gd_3Al_2Ga_3O_{12}$ (GAGG) are an excellent candidate for application in ionizing-radiation detectors because of their high radiation resistance, density and light yield. These crystals can be used in combination with lead tungstate ($PbWO_4$ or PWO) crystals for the development of a new generation of electromagnetic calorimeter with advanced spatial and energy resolutions in a broad energy range. PWO crystals enable the accurate detection of high-energy photons, while GAGG crystals provide the possibility of precisely measuring photon energies, down to a few MeV. Different options for a composite electromagnetic calorimeter based on PWO and GAGG crystals are considered to optimize spatial and energy resolutions in a broad energy range (from 1 MeV to 100 GeV). In particular, different lengths of the GAGG section of the calorimeter are considered, from 0.5 to 10 cm. The separation of signals from photons and hadrons is also taken into consideration through the study of shower shape in the calorimeter. The optimization is based on Geant4 simulations, considering light collection as well as the use of different photodetectors and electronic noise. Simulations are verified with light yield measurements of GAGG samples obtained using radioactive sources and test beam measurements of the prototype of the PWO-based Photon Spectrometer of the ALICE experiment at CERN.

**Keywords:** scintillators; light yield; scintillation detector; calorimetry; high resolution; GAGG; PWO





## 1. Introduction

Homogeneous calorimeters based on scintillation monocrystals are widely used in high-energy physics and related fields due to their excellent energy, spatial and time resolutions. The high-granular photon spectrometer PHOS [1] of the ALICE experiment [2] at the Large Hadron Collider is one example of how such crystals, namely, lead tungstate crystals, $PbWO_4$ (PWO), are used. These crystals are also employed in the CMS experiment [3], as well as in a variety of other experiments in high-energy physics and related fields, such as those conducted on the ISS (the CALET experiment [4]). As shown in [5], such a calorimeter enables one to acquire the resolution of the $\pi^0$-meson peak with a value of $\sigma_m^{\pi^0} = 4.56 \pm 0.03$ MeV/$c$ for $p_\mathrm{T} > 1.7$ GeV/$c$. The good energy and spatial resolution of PWO-based calorimeter allows one to measure experimentally difficult quantities, such as direct photons [6]. However, because of the limited light yield, these measurements are restricted to a relatively high $p_\mathrm{T} \geq 1$ GeV/$c$. Extending measurements to lower energies will allow for the exploration of direct photons in new regions, and even access to new directions, such as tests of the Low theorem [7–9].

For a precise measurement of soft photon energy, crystals with a high scintillation light yield, such as the new promising material $Gd_3Al_2Ga_3O_{12}$:Ce (GAGG), can be employed. The GAGG crystals of different lengths of 0.5, 1.5, and 3 cm (Figure 1) examined in this work were produced by Fomos Materials Company [10]. In comparison to other scintillation monocrystals, GAGG crystals have a relatively good radiation resistance, density, and light yield ([11–13], see Table 1).

GAGG crystals are considered for use in experiments on hadron colliders; for example, in the LHCb calorimeter system after the Phase II upgrade [14]. An operating shashlik calorimeter consists of non-radiation hard scintillator and plastic light guides. It provides an excellent energy resolution; however, in its current form, it cannot be operated after the High-Luminosity LHC Upgrade due to material not being able to sustain such a high level of radiation. The other reasons for the upgrade are the wish to improve the timing resolution and the need to increase granularity while keeping energy resolution at the present level. Possible variants to the upgrade are homogeneous crystals, shashlik module or spaghetti module (SpaCal). However, homogeneous crystals require a long length to contain 25 $X_0$ and have a high cost. Furthermore, no radiation-hard WLS fibers to transport light have been constructed to date for a shashlik-type calorimeter. A spaghetti module can be made very compact and the fibers of the module scintillate and transport light. As GAGG crystals have superb radiation-hardness and a high light yield, they make a perfect candidate for application in SpaCal technology.

The idea of creating a calorimeter with more than one crystal per cell is discussed regarding some future experiments in high-energy physics. For example, the Novel High-Granularity Crystal Electromagnetic Calorimeter for CEPC considers an option of several short crystal bars arranged in both longitudinal and transverse directions with a single-ended readout. The BGO material is considered for a calorimeter that provides a great performance in terms of particle flow analysis, achieving an energy resolution of about $2$–$3\%/\sqrt{E(GeV)}$ [15]. A similar approach is discussed for space-based experiments, introducing the concept of the CaloCube 3D highly-segmented calorimeter [16]. In this calorimeter, an array of $20 \times 20 \times 20$ CsI(Tl) crystals of 3.6 cm side are arranged in a cube; each crystal is equipped with two photodiodes: one for small signals and the other for large signals. This detector is expected to measure cosmic rays in a wide range, from ~10 MeV up to ~100 TeV. A study of different materials for this calorimeter was performed, which considered materials such as CsI(Tl), BaF2, YAP(Yb), BGO, and LYSO(Ce) crystals.

In this work, we consider an electromagnetic calorimeter concept, taking the benefits of a new material with a high light yield—GAGG(Ce)—and a longitudinally segmented calorimeter; however, considering the cost of such a detector, we use only two crystals for a cell: a short GAGG section for low-energy regions, and a longer PWO section to access the energies of several GeVs. We discuss the performance of such a calorimeter in Monte-Carlo simulations. Energy and spatial resolutions are calculated for different GAGG section lengths in order to optimize the cost of such a detector. The separation of signals from photons and hadrons is discussed because high-purity photons are crucial for the possible physical tasks of such a detector.

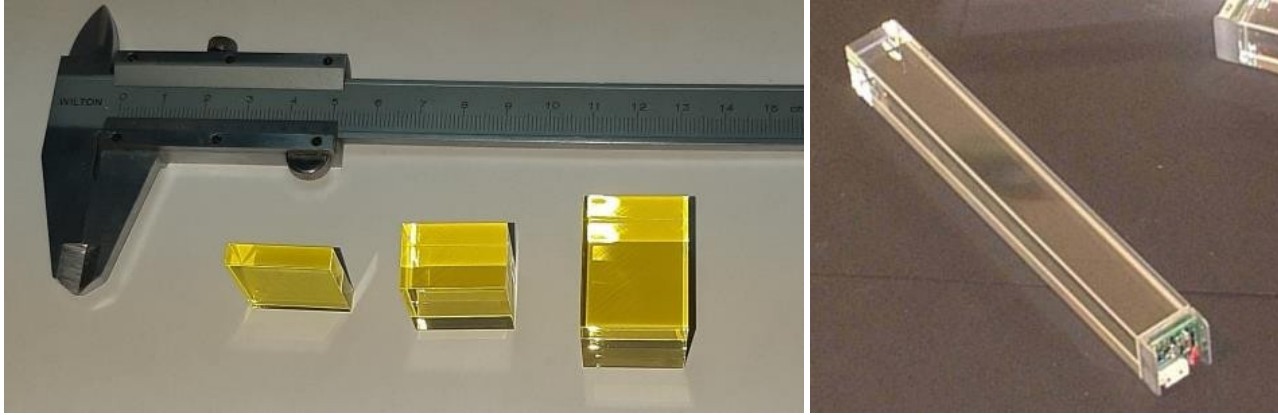

**Figure 1.** (**Left**) GAGG crystals of various lengths. (**Right**) A PWO crystal with a photodetector and a preamplifier attached to it.

**Table 1.** Properties of inorganic scintillation crystals.

| Property | $Gd_3Al_2Ga_3O_{12}$:Ce (GAGG) | $PbWO_4$ (PWO) | $Lu_3Al_5O_{12}$:Ce (LuAG) | $Lu_2SiO_5$:Ce (LSO) | NaI:Tl | CsI:Tl |
|---|---|---|---|---|---|---|
| Density, g/cm$^3$ | 6.68 | 8.28 | 6.73 | 7.4 | 3.67 | 4.53 |
| Wavelength of radiation, nm | 530 | 520 | 535 | 420 | 415 | 550 |
| $X_0$, cm | 1.59 | 0.89 | 1.30 | 1.10 | 2.60 | 1.86 |
| Light yield, photons/MeV | 50,000 | 100–300 | 25,000 | 30,000 | 40,000 | 54,000 |
| Decay time, ns | 95 | 20 | 70 | 70 | 230 | 680 |
| Hygroscopicity | None | None | None | None | Strong | Slight |

## 2. Simulations in Geant4

A computer program based on the Geant4 [17] package was developed to determine the energy resolution and other features of electromagnetic calorimeter models. The program allows for calculations in the case of both a calorimeter with only one type of crystal and the case of a compound calorimeter with GAGG and PWO crystals arranged in series. The simulation uses a calorimeter assembly of 11 × 11 crystals in a steel honeycomb structure with a wall thickness of 0.01 cm. In the case of the compound calorimeter, GAGG crystals of the following lengths were examined: 0.5, 1.5, 3, 5, and 10 cm (see example layouts in Figure 2); the length of the PWO section was 18 cm (such crystals are used in the PHOS calorimeter). The cross-section of the crystals in both cases was 22 × 22 mm$^2$. The energy released in each section was calculated, then recalculated into the photodetector signal according to the light yield value. Separate photodetectors were proposed for each segment. The primary particle's entry point was evenly distributed over the surface of the calorimeter's central crystal. Photons with energy levels ranging from 100 keV to 100 GeV were examined.

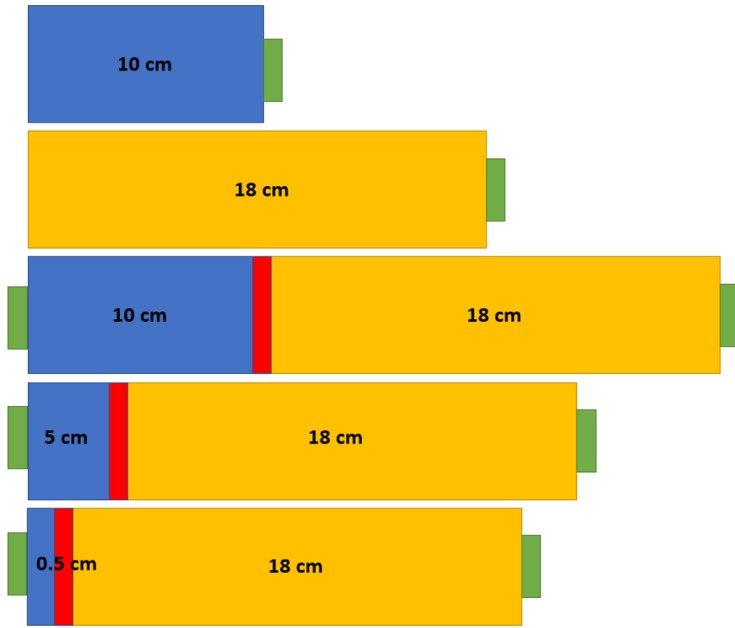

**Figure 2.** Schematic variants of GAGG and PWO crystal arrangements with photodetectors: green—photodetector, red—insulator, blue—GAGG crystal, yellow—PWO crystal.

An optional value of the deposited energy in each crystal was modified to provide a more realistic description of the calorimeter:

(1) Light yield simulation: the number of photons on the photodetector changes according to the Poisson distribution, with mean values of 420 phe/MeV and 6 phe/MeV for

GAGG crystals and PWO crystals in the case of APD 5 × 5 mm$^2$ at T = –25 °C (for a convenient comparison with test beam measurements using PHOS [18]), respectively, and then recalculated back into energy;

(2) Emulation of the noise of electronics, which was modeled as a random Gaussian distribution with parameters $\mu = 0$ and $\sigma = 5$ MeV (high noise) or $\sigma = 1$ MeV (low noise) and then added to the energy released in each cell;

(3) Cuts to the minimum energy in a cell: only those cells with an energy release above some set threshold (10 MeV for high noise; 2 MeV for low noise) were included;

(4) Clustering: as in the ALICE analysis and simulation package, AliROOT [2], a cell was discarded if it did not have at least one vertex in common with the rest of the cluster.

### 2.1. Validation of the Simulations for Pure GAGG and PWO Calorimeters

Gamma spectra of $^{22}$Na and $^{137}$Cs radioactive sources were measured in GAGG crystals of different lengths. The measurements were conducted using a PMT-143 photomultiplier with a quantum efficiency of 17.6% [19]. The results of the measurements and approximations can be found in a previously published article [20]. The calculated weighted average value of the light yield was 420 ± 16 phe/MeV. This value was further utilized in simulations to characterize the light yield of any length of GAGG crystal.

The simulation of the response of a single crystal reproduces the experimental energy spectra of GAGG crystals obtained using radioactive sources (see Figure 3; crystal length 3 cm). The small deviation in the position of the second and third peaks can be explained by effects not included in the simulation; for instance, light absorption in the crystals.

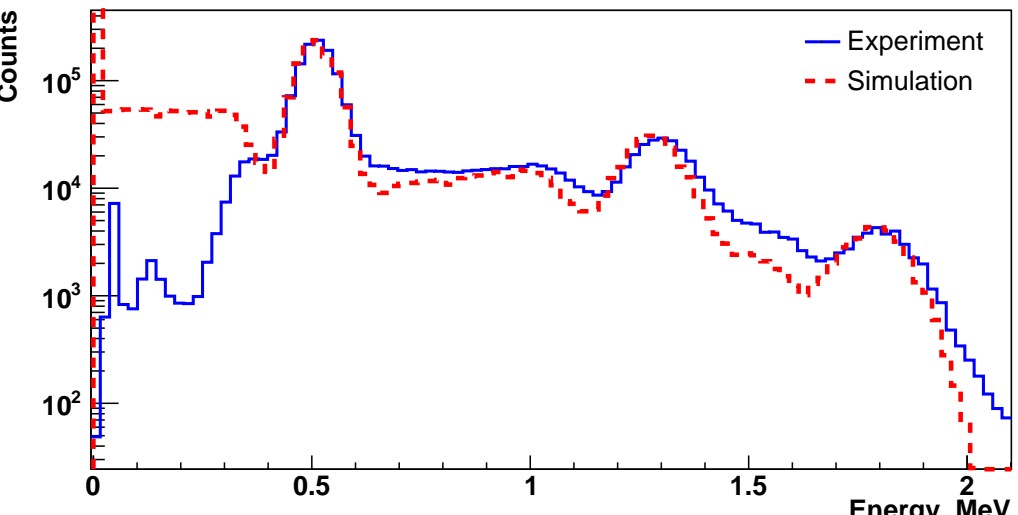

**Figure 3.** An example of the reproduction of experimental data in simulations. GAGG crystal length is 3 cm [20].

To validate the simulation model, the spatial and energy resolutions of the assembly consisting solely of PWO crystals were compared to the experimental data collected with the ALICE experiment's PHOS detector.

To obtain the spatial resolution, a two-dimensional distribution of the distance from the photon's entrance point into the assembly to the cluster's center of gravity was calculated with a logarithmic weight [1] (Figure 4, left). The projection of such a distribution on the x-axis was then plotted (Figure 4, right), and the standard deviation, which represents the spatial resolution, was calculated.

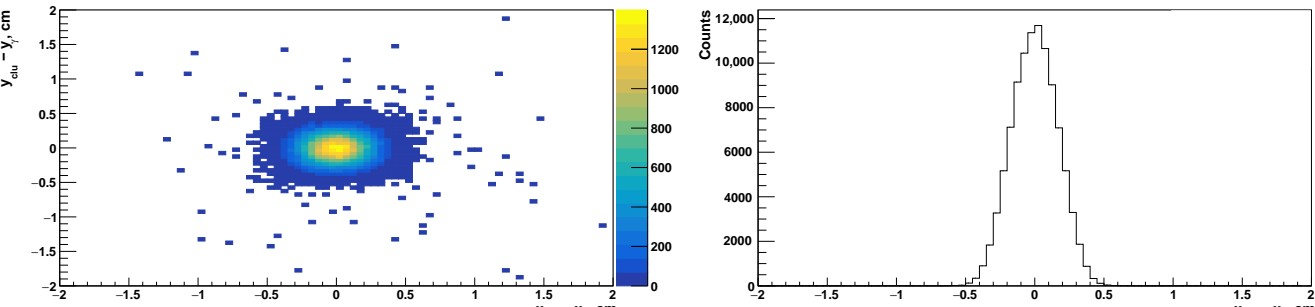

**Figure 4.** (**Left**) An example of a two-dimensional distribution of the coordinates of the cluster center of gravity for $10^5$ photons with an energy of 5 GeV; (**right**) a projection of the two-dimensional distribution on the x-axis [20].

The dependence of the spatial resolution of the assembly consisting of PWO crystals on the energy, with and without the noise being taken into account, is shown in Figure 5. The results of the PHOS detector of the ALICE experiment were used as a reference [1,2].

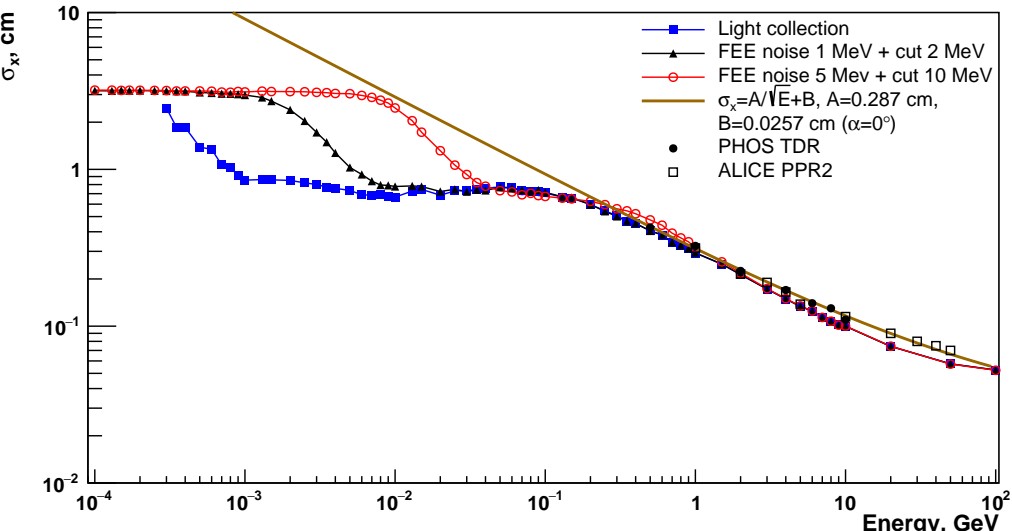

**Figure 5.** Energy dependence of the spatial resolution of PWO crystals in the simulation compared to experimental results [20].

One can see that the spatial resolution obtained in the simulations reproduces the experimental data rather well, especially when the 5 MeV noise and the 10 MeV minimum cell energy cut are included.

To calculate the energy resolution of the PWO-calorimeter, the obtained distributions of the deposited energy were approximated by the Gaussian function and its parameters $\sigma$ and $E_{mean}$ were determined.

The obtained dependence of the energy resolution on the energy of the primary photons is shown in Figure 6. The experimental curve obtained in beam tests of the PHOS prototype is given as a reference [18]. One can see that the simulated resolution and experimental resolution are almost the same in the case of high noise, but the simulation generally offers a better resolution. This can be explained by the higher electronic noise level in the experiment. When the noise is reduced, the resolution improves across the entire energy range.

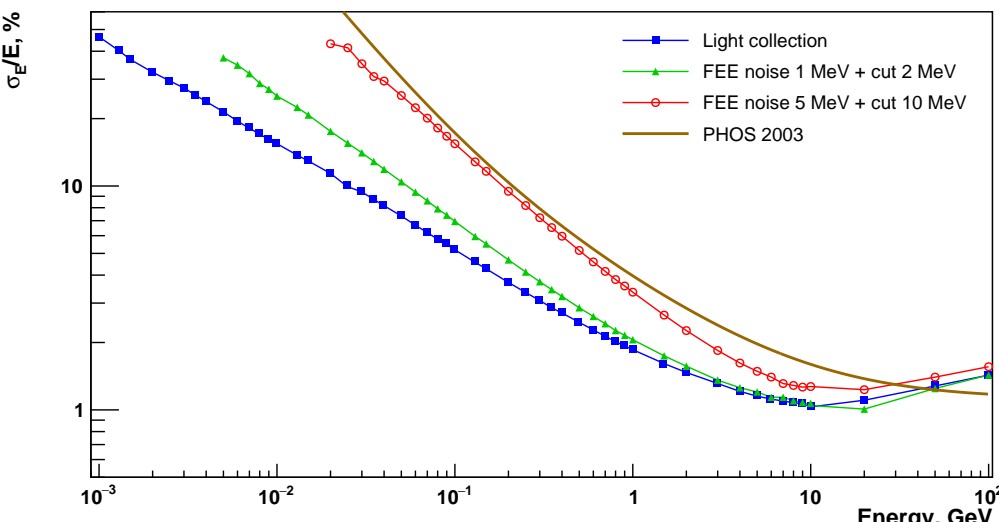

**Figure 6.** Energy resolution of PWO crystals with and without the noise simulation [20].

Thus, the simulation reproduces the spatial and energy resolution of the PWO crystals quite well. Therefore, it is possible to consider a model of a compound calorimeter of sequentially arranged GAGG and PWO crystals and predict the energy and spatial resolutions of the calorimeter based on our model with a sufficient degree of confidence.

## 2.2. Results of the Simulations for Combined Calorimeters

It is crucial to determine whether it is possible to distinguish between photons and other types of particles with the proposed combined GAGG + PWO-calorimeter. There are a few methods that provide such an opportunity. The first one is based on a pulse shape analysis of different particles, for instance, positrons and neutrons [21,22] or photons and $\alpha$-particles [23]. The second way to separate gammas and neutrons considers the difference in their radiation/interaction lengths. This can be achieved by the usage of different crystals such as PWO and GAGG [22], or two identical GAGG:Ce crystals with one of them codoped with Mg [21]. The third option is to discriminate photons by their shower shape and suppress the charged pion and antineutron background. This is achieved by calculating the eigenvalues ($\lambda_{short}$, $\lambda_{long}$) of the dispersion tensor [1]. Examples of such distributions for photon energies of 500 MeV and 5 GeV, obtained by the GAGG + PWO calorimeter with a 10 cm long GAGG section, are depicted in the Figure 7. At high photon energies, electromagnetic showers are rather compact and localized in a narrow region in the plane ($\lambda_{short}$, $\lambda_{long}$). The distribution broadens with a decrease in the photon energy.

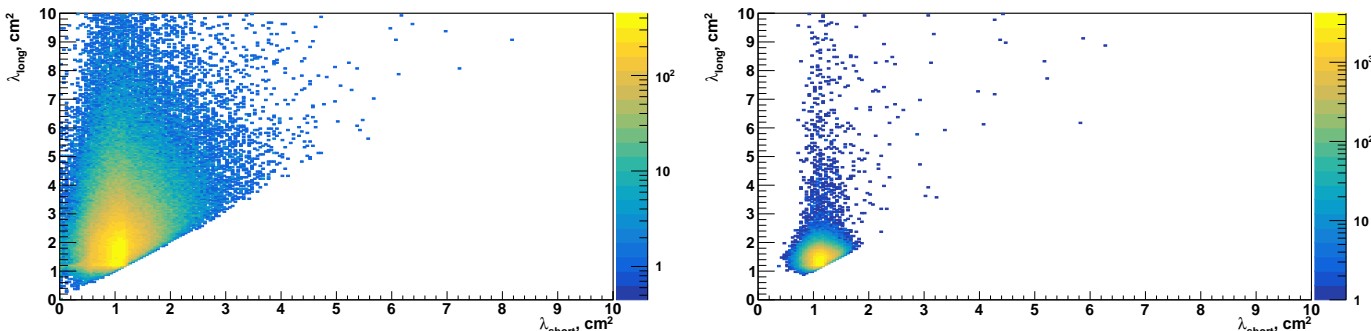

**Figure 7.** Examples of ($\lambda_{short}$, $\lambda_{long}$) distributions for a photon impact into GAGG + PWO-calorimeter with the photon energy of 500 MeV (**left**), 5 GeV (**right**).

Analogous distributions can be constructed for other types of particles, such as charged pions and antineutrons. Projection examples for particles with energies of 5 GeV and

500 MeV are represented in the Figures 8 and 9, respectively. It can be concluded that $\pi^+$ and $\bar{n}$ distributions are much broader than $\gamma$ distributions, which provides an opportunity for an efficient gamma–hadron separation based on a $\lambda$ cut. In addition, the distribution of antineutrons remains broad, regardless of the primary photon energy. After applying a cut ($\lambda_{short} \leq 1.5$ cm$^2$, $\lambda_{long} \leq 2$ cm$^2$) for 5 GeV particles, only 1.8% of $\gamma$ are lost, but 89.9% of $\pi^+$ and 98.7% of $\bar{n}$ are rejected. After applying a cut ($\lambda_{short} \leq 2.2$ cm$^2$, $\lambda_{long} \leq 5$ cm$^2$) for 500 MeV particles, only 7.2% of $\gamma$ are lost, but 55.9% of $\pi^+$ and 96.2% of $\bar{n}$ are removed.

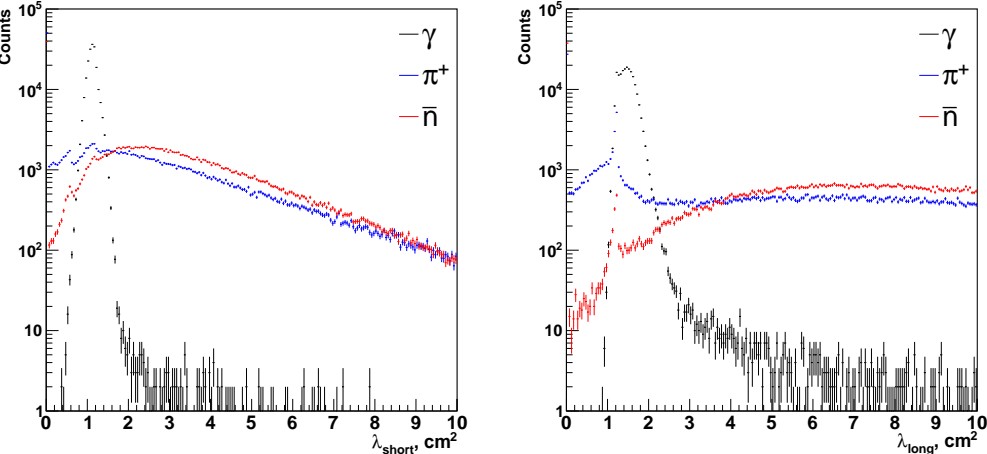

**Figure 8.** Projections of ($\lambda_{short}$, $\lambda_{long}$) distribution for 5 GeV particles ($\pi^+$, $\bar{n}$, $\gamma$) on the: x-axis (**left**), y-axis (**right**).

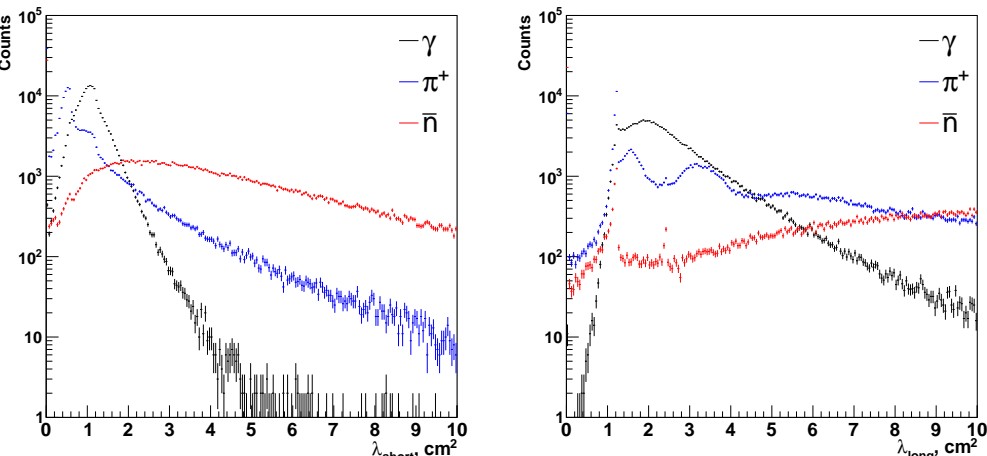

**Figure 9.** Projections of ($\lambda_{short}$, $\lambda_{long}$) distribution for 500 MeV particles ($\pi^+$, $\bar{n}$, $\gamma$) on the: x-axis (**left**), y-axis (**right**).

The level of possible $\gamma/\pi^+$ and $\gamma/\bar{n}$ differentiation is shown in Figure 10. Here, the accepted fraction of particles with $\lambda_{long} > 0$ and a less than varying cut as a function of this cut is shown. At the same time, $\lambda_{short}$ is fixed at the value of 2.2 cm$^2$. A $\lambda$ cut, which depends on the deposited energy, can be used to obtain an energy-independent acceptance level for electromagnetic showers while providing a maximum hadronic shower rejection.

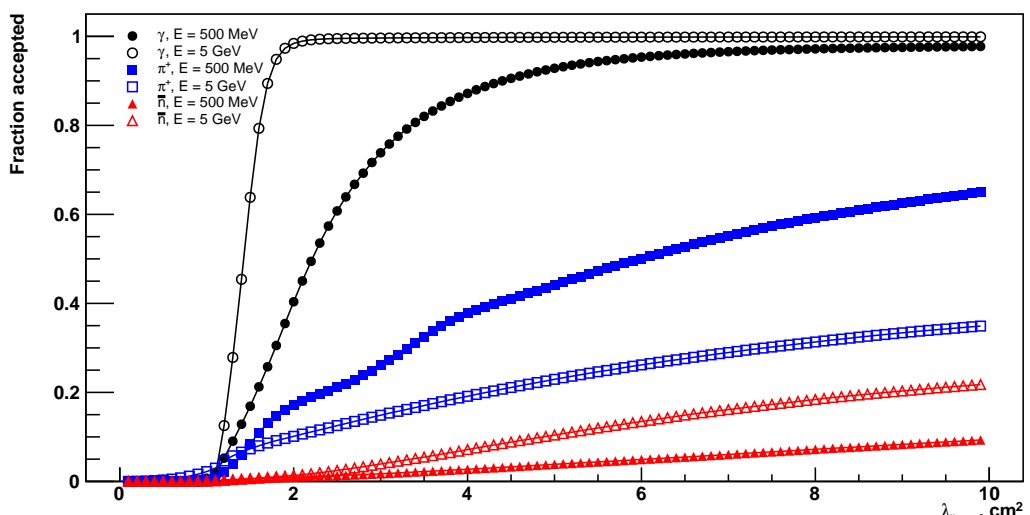

**Figure 10.** The fraction of gammas and hadrons accepted for varying upper limit on the shower dispersion as a function of this cut with fixed $\lambda_{short}$ = 2.2 cm$^2$.

To determine the spatial resolution of the GAGG and GAGG + PWO calorimeters, the procedure described above for the calculation of PWO calorimeter resolution was carried out. Figure 11 shows a comparison of the spatial resolution in the PWO-, GAGG-, and GAGG + PWO-calorimeters, excluding noise. The spatial resolution of the combined calorimeter is slightly worse than that of the calorimeter consisting only of PWO crystals, but not significantly. The resolution of the pure GAGG-calorimeter is noticeably worse in the range of energies of 1–100 MeV and above 5 GeV.

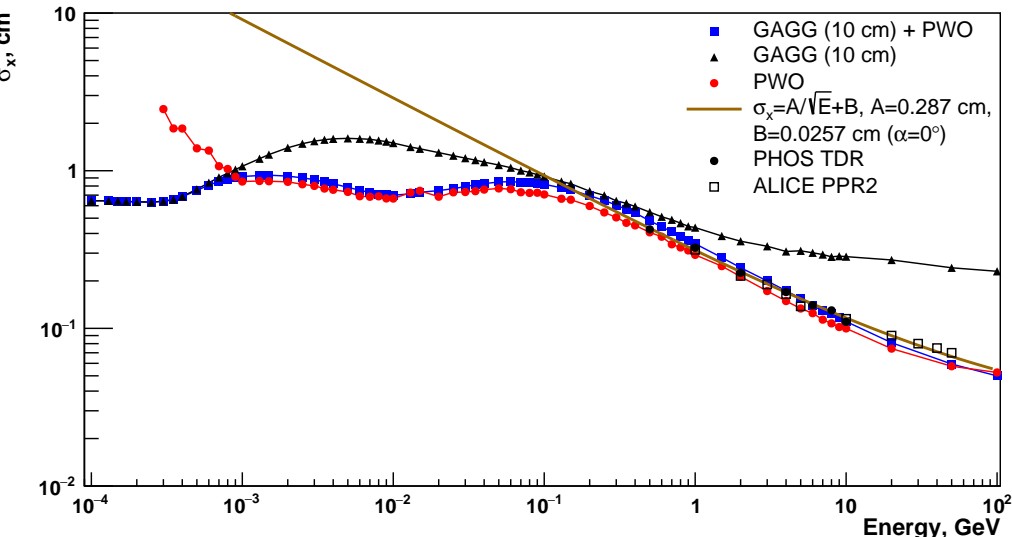

**Figure 11.** Comparison of spatial resolution in PWO-, GAGG-, and GAGG + PWO-calorimeters, excluding noise [20]. Experimental data are taken as a reference (PHOS).

A comparison of the spatial resolution obtained for GAGG + PWO-calorimeter with different GAGG-section lengths is depicted in Figure 12. For the energies above 1 MeV, the spatial resolution of the GAGG + PWO-calorimeter is nearly the same as the resolution of the PWO-calorimeter. However, for the energies below 1 MeV, the situation is inverse: the spatial resolution of GAGG + PWO improves by 2–4 times.

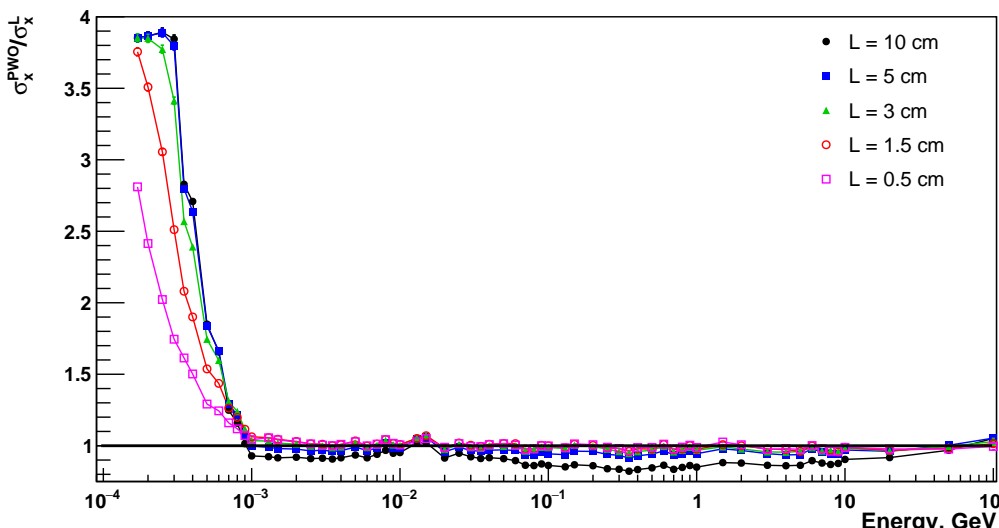

**Figure 12.** The ratio of the spatial resolution of PWO-calorimeter to the one of GAGG + PWO-calorimeters with different GAGG-section lengths (excluding noise).

In the case of a GAGG calorimeter or a combined GAGG + PWO calorimeter, the energy distributions are essentially non-Gaussian, so it seems more correct to approximate them with the Crystal Ball function [24] (which better describes the decaying "tails" in the distributions):

$$f(x) = \begin{cases} e^{-\frac{(E-E_{mean})^2}{2\sigma^2}}, & \frac{E-E_{mean}}{\sigma} > -\alpha \\ A \cdot \left(B - \frac{E-E_{mean}}{\sigma}\right)^{-n} + C, & \frac{E-E_{mean}}{\sigma} \leq -\alpha \end{cases} \tag{1}$$

From here, by obtaining parameters $\sigma$ and $E_{mean}$, one can construct the energy resolution. Figure 13 shows such a resolution for the case including light collection and excluding the noise in the PWO-calorimeter, as well as in the GAGG- and GAGG + PWO-calorimeters with 3 cm long GAGG crystals. Experimental data obtained from radioactive sources are also included as a reference. The simulated resolution and experimental resolution of the GAGG-calorimeter are nearly the same; thus, once again, the correctness of the simulations is proven. This difference can be explained by the fact that, in a simulation for the light yield, the mean value of 420 phe/MeV is used but is expected to vary for crystals of different lengths due to the light absorption. Furthermore, for the GAGG-calorimeter, it is not possible to obtain the energy resolution in the whole range of photon energies (from 100 keV to 100 GeV) as, with the energy increase, the peak in energy distribution disappears or becomes non-fittable with the Gaussian-type distribution. The first sharp increase in the energy resolution in the range of 60–100 MeV can be explained by the increase in the energy deposited in PWO-section; the second one occurs due to the limited length of GAGG crystals. On the other side, at low energies,, the resolution of the GAGG-calorimeter is an order of magnitude better than that of the PWO-calorimeter: ∼5% instead of ∼45% at $E$ = 1 MeV. In the case of the compound calorimeter (GAGG + PWO), the energy resolution is better than that in the PWO calorimeter across the entire energy range (from 100 keV to 100 GeV).

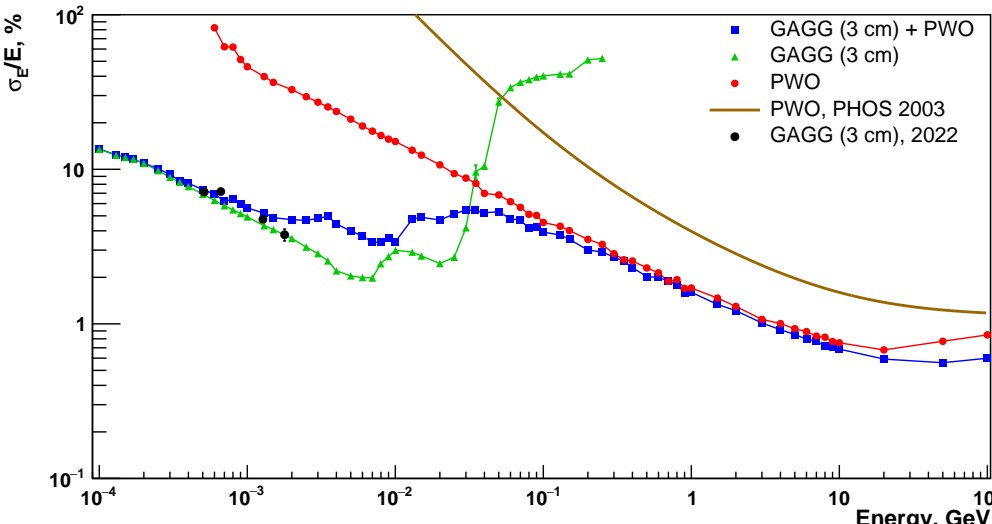

**Figure 13.** Comparison of the energy resolution in PWO-, GAGG-, and GAGG + PWO-calorimeters excluding noise with a GAGG-section length of 3.0 cm.

A comparison of the energy resolution obtained for GAGG + PWO-calorimeter with different GAGG-section length is depicted in the Figure 14. Below 10 MeV, the energy resolution of the compound calorimeter becomes 1.5–10 times better than the resolution of the PWO-calorimeter. This improvement is also seen for energies above 20 GeV. As one can see from the Figure 15, the optimal length of the GAGG-section is about 3 cm because further GAGG-section length increases do not result in the energy resolution becoming significantly better, although the cost of the crystals rises.

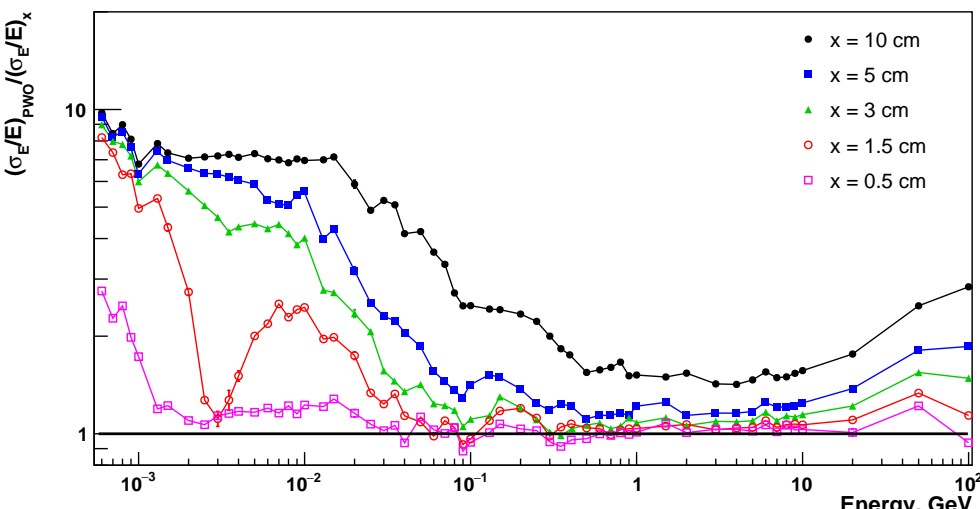

**Figure 14.** The ratio of the energy resolution of PWO-calorimeter to that of GAGG + PWO-calorimeters with different GAGG-section lengths (excluding noise).

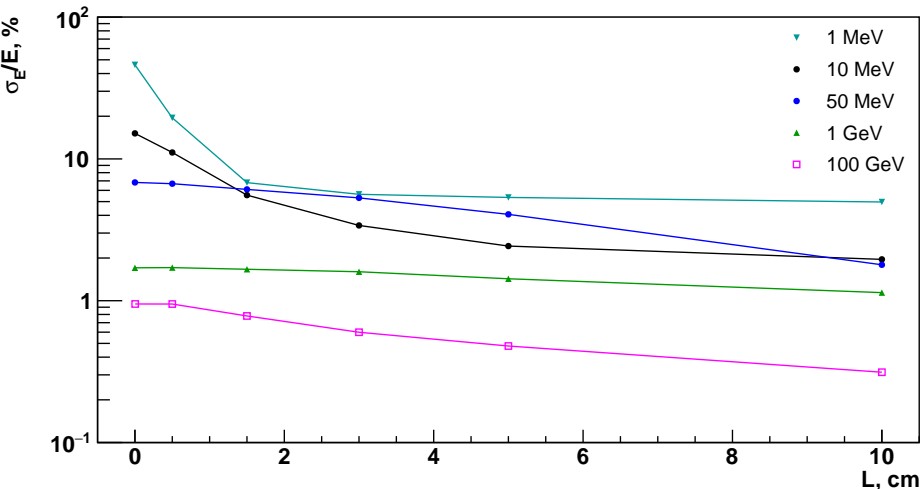

**Figure 15.** The dependence of the energy resolution on the GAGG-section length of the GAGG + PWO-calorimeter for photons of different energies.

### 3. Conclusions

We performed a simulation of an electromagnetic calorimeter based on scintillation crystals PWO, GAGG, and the composite version GAGG + PWO. It was shown that the composite calorimeter has approximately the same energy resolution as GAGG based at low energies and a somewhat better energy resolution than PWO based at high energies. The GAGG + PWO calorimeter has nearly the same spatial resolution as the PWO-calorimeter. A combined calorimeter of this type will enable the measurement of photon energy in an unprecedentedly broad range of energies. A combined calorimeter will also allow for particle identification based on the longitudinal and transverse shape of the shower. The results regarding the efficiency and purity of $\lambda$ cuts in the case of the GAGG + PWO calorimeter with a 10 cm GAGG section are summarized in Table 2. For lower lengths of the GAGG section, $\gamma$ efficiency stays almost the same, but hadron contamination becomes worse, by $\approx 20\%$ for $\pi^+$ and $\approx 40\%$ for $\bar{n}$ for the 3 cm GAGG section.

**Table 2.** Efficiency and purity after applying dispersion cuts.

| Cut | Energy | $\gamma$ Efficiency, % | Contamination $\pi^+$, % | Contamination $\bar{n}$, % |
|---|---|---|---|---|
| $(\lambda_{short} \leq 1.5\ \text{cm}^2, \lambda_{long} \leq 2\ \text{cm}^2)$ | 5 GeV | 98.2 | 10.1 | 1.3 |
| $(\lambda_{short} \leq 2.2\ \text{cm}^2, \lambda_{long} \leq 5\ \text{cm}^2)$ | 5 GeV | 99.8 | 22.9 | 10.4 |
| $(\lambda_{short} \leq 1.5\ \text{cm}^2, \lambda_{long} \leq 2\ \text{cm}^2)$ | 500 MeV | 39.5 | 17.1 | 0.9 |
| $(\lambda_{short} \leq 2.2\ \text{cm}^2, \lambda_{long} \leq 5\ \text{cm}^2)$ | 500 MeV | 92.8 | 44.1 | 3.8 |

The optimization of the length of the GAGG section of the GAGG+PWO calorimeter cell is performed on the basis of the energy resolution improvements in different energy ranges for lengths from 0.5 to 10 cm. It was shown that, for 10 MeV photons, the dependence on the GAGG-section length is most prominent, and the optimal value is about 3 cm.

**Author Contributions:** Conceptualization , D.B.; Software, D.A.; Formal analysis, D.A.; Writing—original draft, D.A.; Writing—review & editing, D.B.; Visualization, D.A.; Supervision, D.B. All authors have read and agreed to the published version of the manuscript.

**Funding:** The research was carried out with the financial support of the Russian Science Foundation, grant No. 22-42-04405. The electromagnetic calorimeter modeling program was developed with the support of the National Research Center Kurchatov Institute.

**Institutional Review Board Statement:** Not applicable.

**Informed Consent Statement:** Not applicable.

**Data Availability Statement:** Not applicable.

**Acknowledgments:** The authors express their gratitude to D. Y. Peresunko, M. S. Ippolitov, and Y. V. Kharlov, the staff of the Quark Matter Laboratory at the National Research Center Kurchatov Institute, as well as to M. V. Korzhik, G. A. Dosovitskii, and P. V. Karpyuk, the staff of the Institute of Chemical Reagents and High Purity Chemical Substances of the National Research Centre Kurchatov Institute, for their help in obtaining experimental data, discussing the results, and remarks on the text.

**Conflicts of Interest:** The authors declare no conflict of interest. The funders had no role in the design of the study; in the collection, analyses, or interpretation of data; in the writing of the manuscript; or in the decision to publish the results.

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
