# Peer review of "New Inogranic Scintillators’ Application in the Electromagnetic Calorimetry in High-Energy Physics"

_applsci, doi:10.3390/app13106189_

Round 1

Reviewer 1 Report

Referee comments on the paper: 

"New inorganic scintillators application for the electromagnetic calorimetry in high-energy physics”  by Dmitry Averyanov and Dmitry Blau, submitted to Applied Physics, MDPI.

Calorimeters using scintillation monocrystals providing high energy, spatial, and time resolution are widely used in high-energy experiments on modern colliders.

In the review paper, optimization of the detectors based on the PWO scintillation crystals for optimization of the high-energy gamma ray detection to obtain precise energy estimation in the energy range of few MeV and higher.

Optimization was done with the GEANT4 package and checked with test beam measurements with Alice’s experiment photon spectrometer prototype. The simulation reproduces the spatial and energy resolution of the PWO crystals quite well.

The next major problem for high-energy physics experiments is particle classification based on the shape of the energy release of different particles or different attenuation lengths. Also, discrimination can be made by examining the shower shapes initiated by different particles in the calorimeter. The photon showers are much more compact compared with ones initiated by charged particles. From the two or more-dimensional distributions of particle parameters, it is possible to outline the best decision surface to keep the photon detection efficiency high enough and simultaneously high suppress the contamination of alternative particle contamination.

Usually, it was made by the liner cuts method in the parametric space; the neural neuron network or Bayesian classifiers now substitute it.

The authors conclude, "The GAGG + PWO calorimeter will enable the measurement of photon energy in an unprecedentedly broad range of energies. A combined calorimeter will also allow for particle identification based on the longitudinal and transverse shape of the shower.

Authors should give in the conclusion exact numerical estimates of achieved accuracies and classification results in the purity-efficiency table. 

With this necessary addition, I'll recommend the paper for publication as it is.

Author Response

Dear Sir/Madam,

Thank you for high acclaim of our work. We estimated gamma purity and hadron contamination and included exact values in the table in conclusions, as you suggested. Purity-efficiency charts in different environments (proton-proton, heavy-ion collisions) at different colliding energy, detector layouts and distances require detailed studies and will be reported in future publications.

Best regards,

authors

Reviewer 2 Report

The study presented in this article is very interesting and well positioned on the cutting edge of the calorimetry in high energy physics.

A simulation analysis of the energy and spatial resolution for different calorimeters based on different arrangements of GAGG and GAGG + PWO scintillators are presented. 

Nevertheless, the authors published few months ago a very similar paper on Physics of Atomic Nuclei, 2023, Vol. 86, No. 1, pp. 33–43, entitled "Perspectives of Inorganic Scintillator GAGG Application for Precision Electromagnetic Calorimetry", having as DOI: 10.1134/S1063778823010052.

As can be easily checked, the abstract and the text up to line 137 of the proposed draft is almost equal to the article already published, and all the pictures and the tables from 1 to 8 are the same, except for minor editing differences. Also fig. 12 and fig. 14 have been already published in the same article. The 90% of the conclusions are the same.

The original part of the paper, regarding the analysis of the longitudinal and transversal shape of the shower and the analysis of the spatial and energy resolution of different calorimeters based on different arrangements of GAGG scintillators having different lengths, is just a deeper argumentation about already published results. 

I suggest writing a different manuscript, giving more space to the interesting original results proposed, and reducing the common part to the article already published in Physics of Atomic Nuclei. In particular, it is necessary to summarize the introductory part and reduce the number of figures describing the results of the calibration measurements of the GAGG crystals. Figures already published can be reused, if necessary, quoting the reference.

Author Response

Dear Sir/Madam,

Thank you for high acclaim of our work. As you suggested, we prepared a deeper introduction part and removed the part about calibration and quoted results from the previous work which are needed as a basis and verification for the current work.

We prepared more details about gamma-hadron separation with an additional figure which was also demanded by another reviewer.

Best regards,

authors

Round 2

Reviewer 2 Report

This second version of the manuscript is almost fine, except for few points that needs to be changed or modified to improve the overall quality of the paper.

1) The introduction needs to be improved to better clarify the statements about the proposed new calorimeters for the LHCb.

Please rephrase the following parts: line 39-41; line 45-48.

---------------------------------------------------

2) In this version of the manuscript Section 2 is useless and can be eliminated. The only important outcome of this section (the measured light-yield) can be reported in the next section. I suggest, as an alternative, to add to this section a very short summary of the results obtained with the measurements done using radioactive source.

-----------------------------------------------------------------

3) To my opinion section 3 is not well organized. I mean that several results are presented in this order, one after the other:

a) Energy spectra data-simulation comparison for GAGG Crystal

b) Spatial resolution for PWO calorimeter

b) Energy resolution for PWO calorimeter 

d) Particle discrimination (gamma, anti-neutron, pi+) with a combined calorimeter GAGG+PWO (10cm) 

e) Spatial resolution for combined calorimeter GAGG+PWO

f) Energy resolution for combined calorimeter GAGG+PWO

As it is, is not simple for the reader to follow the informations in the text and the methodology followed  is not clearly stated.

To avoid this I suggest to divide the section n.3 in two subsection as following:

1) Validation of the simulation for pure GAGG and PWO calorimeters

2) Results of the simulations for combined calorimeters

thus underlining in a stronger way what you said in the text: you are able to reproduce the performances of both single crystal calorimeters before you analyze the case of a compound calorimeter.

As an alternative, you can organize the sections dividing the text (and consequently the results) by subject: I mean putting together all the results about spatial resolution, then energy resolution ones, etc. without spreading information about the same physical quantity all over the text. 

--------------------------------------------------------

I have one more questions: does the size of the GAGG part in the compound calorimeters affect the particle discrimination performances? In the text (and in table 2) you show the results obtained for a 10 cm GAGG compound calorimeter, but you also state that the optimal length for the GAGG-section is about 3 cm because further GAGG-section length increase does not improve too much the energy resolution. What can you say on the particle identification with a 3 cm length for the GAGG-section?

Author Response

Dear Sir/Madam,

  Thank you for comments and suggestions. 1. It is not very clear what should be clarified, but we tried to rephrase this text. 2-3. Text changed as you suggested: section 2 removed and text moved to new subsection 2.1 "Validation". Results for combined GAGG+PWO calorimeter moved to subsection 2.2. 4. We checked particle identification for a calorimeter with 3 cm GAGG section. You can find results in the attached pdf file. We added a short summary about it in the conclusion section, not including full table. If you think we should include the table as well, we can do it.   Best regards,  

authors
